# DeFusion: An Effective Decoupling Fusion Network for Multi-Modal Pregnancy Prediction

**Xueqiang Ouyang**[1]               202221044618@mail.scut.edu.cn
**Jia Wei**[*1]                     csjwei@scut.edu.cn
**Wenjie Huo**[2]                   1282039293@qq.com
**Xiaocong Wang**[2]                 xwang@smu.edu.cn
**Rui Li**[3]                      rxlics@rit.edu
**Jianlong Zhou**[4]               jianlong.zhou@uts.edu.au

[1] *School of Computer Science and Engineering, South China University of Technology, Guangzhou, China*

[2] *Department of Obstetrics and Gynecology, Nanfang Hospital, Southern Medical University, Guangzhou, China*

[3] *Golisano College of Computing and Information Sciences, Rochester Institute of Technology, Rochester, NY 14623, USA*

[4] *UTS Data Science Institute, University of Technology Sydney, Ultimo, NSW 2007, Australia*

**Editors:** Accepted for publication at MIDL 2025

## Abstract

Temporal embryo images and parental fertility table indicators are both valuable for pregnancy prediction in **in vitro fertilization embryo transfer** (IVF-ET). However, current machine learning models cannot make full use of the complementary information between the two modalities to improve pregnancy prediction performance. In this paper, we propose a Decoupling Fusion Network called DeFusion to effectively integrate the multi-modal information for IVF-ET pregnancy prediction. Specifically, we propose a decoupling fusion module that decouples the information from the different modalities into related and unrelated information, thereby achieving a more delicate fusion. And we fuse temporal embryo images with a spatial-temporal position encoding, and extract fertility table indicator information with a table transformer. To evaluate the effectiveness of our model, we use a new dataset including 4046 cases collected from Southern Medical University. The experiments show that our model outperforms state-of-the-art methods. Meanwhile, the performance on the eye disease prediction dataset reflects the model's good generalization. Our code and dataset are available at https://github.com/Ou-Young-1999/DFNet.

**Keywords:** decoupling fusion, multi-modal fusion, IVF-ET pregnancy prediction.

## 1. Introduction

Recent study shows that up to 12-15% of couples are diagnosed as infertility (Hornstein, 2016), and **in vitro fertilization embryo transfer** (IVF-ET) is one of the most effective technologies to treat infertility. As shown in Fig. 1, during the IVF-ET process, medical laboratory technicians obtain multiple oocytes by stimulating mother's uterus with ovulation and produce multiple zygotes in a laboratory environment (Hanevik and Hessen,

---

* Corresponding author: csjwei@scut.edu.cn

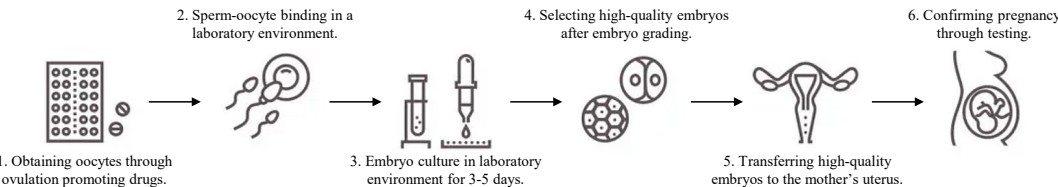

Figure 1: The committed step of IVF-ET.

2022). After 3-5 days culture, laboratory technicians select the optimal embryos based on visual evaluation of embryo morphology and transfer it back to mother's uterus for further development. Thus, it is a crucial step to select high-quality embryos that would lead to promising pregnancy results of IVF-ET.

In clinical practice, the pregnancy success rate of IVF-ET is 30-40% only (Gleicher et al., 2019). One reason is that the optimal embryos may not be survival after the transfer, since human evaluation of embryo morphology is highly subjective and with low consistency. Moreover, embryo morphology is not always relevant to embryos' true development vitality. Numerous studies have shown that embryos with good morphology didn't survive while ones with poor morphology did (del Carmen Nogales et al., 2021). In fact, not only the embryo morphology affects the pregnancy success rate, but also the fertility indicators of parents, such as parents' age, endometrial thickness, sperm quality, and so on.

In the area of computer assisted IVF-ET, existing researches mainly focus on the morphological grading of embryos. As shown in Fig. 1, we can obtain microscope images of embryonic development between step 3 and step 4 of IVF-ET. In order to perform the embryos morphological grading task, (Wu et al., 2021) and (Liu et al., 2023a) apply convolutional neural networks (CNN) based on static embryo images; (Wang et al., 2024) and (Lukyanenko et al., 2021) apply transformer and two-stream neural network based on time-lapse microscopy (TLM) images, respectively. In addition, (Cheng et al., 2024) fuse multi-focal images to predict grade of blastocyst. The performance of these methods outperform laboratory technicians, because embryos grading is completely based on morphological information and the salient morphological characteristics among different grades are easily distinguishable for machines. However, morphological grading is indirect and less relevant to the pregnancy outcome of IVF-ET as discussed in the previous paragraph. Therefore, more researches tend to predict the pregnancy outcome directly.

Recent AI-based assessment models achieve promising success in direct pregnancy predction. (Dehghan et al., 2024) apply traditional machine learning methods with fertility table indicators obtained before step 5 for pregnancy prediction. In addition, static images of the fifth day's embryos and TLM images are also adopted to predict the pregnancy outcome, respectively (Kim et al., 2024; Berntsen et al., 2022). What's more, (Liu et al., 2022)(**MMBE**) fuse the fifth day's static embryo image and fertility table indicators to achieve better pregnancy prediction performance. The major limitation of the existing image-based methods is that they are only applicable to the fifth day's embryo transfer. However, in reality many reproductive centers carry out the third day's embryo to transfer. Although the embryo of the third day is less developed than that of the fifth day, the embryo images of the third day can still provide clinically significant information for pregnancy

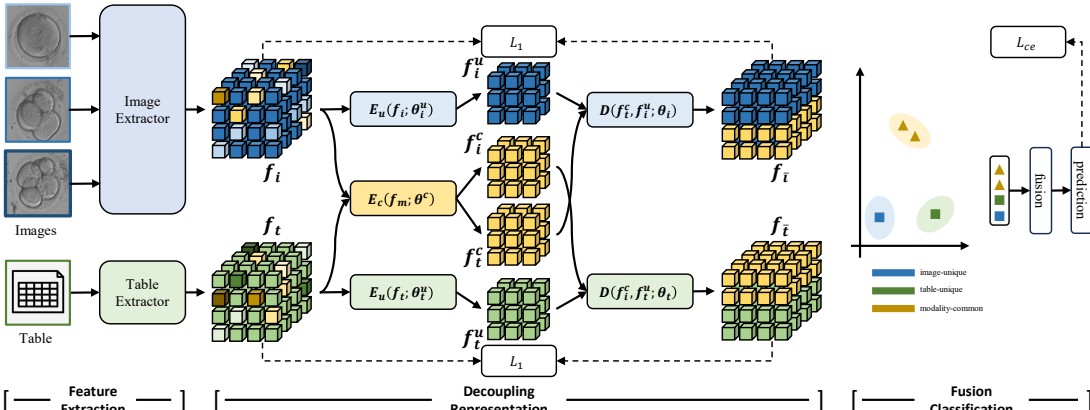

Figure 2: The framework of DeFusion. In the process of decoupling training, the features of different modalities change from entangled to disentangled.

prediction (Neblett et al., 2021). On the other hand, due to some technical constraints in reality multi-modal fusion method (Liu et al., 2022) can only be used to the last day's image.

To address the limitations discussed above, we propose a Decoupling Fusion Network called DeFusion to effectively integrate temporal images of the first three days and parental fertility table indicators for IVF-ET pregnancy prediction. The main contributions are summarized as follows:

- DeFusion is the first to integrate the first three days of embryonic development temporal images and parental fertility table indicators for pregnancy prediction.

- We propose a spatial-temporal position encoding for fusing temporal embryo images. Moreover, we apply a table transformer to extract tabular information from fertility indicators.

- We propose a novel decoupling fusion network to fuse multi-modal information more finely grained by decoupling information from different modalities into modality related and unrelated feature.

## 2. Method

In the field of medical multi-modal fusion, the final fusion features are obtained by complementing the unique features and enhancing the common features of different modalities. However, there is a complex relationship between the features of the same modality and different modalities, which is not a simple linear relationship. So it is difficult to be captured by the model. Inspired by the decoupling operation in (Dong et al., 2023) (Li et al., 2023a), we use the decoupling fusion strategy explicitly decouples the features of different modalities into unique and common features, which is a shift from entangled features to disentangled ones, simplifying the relationships between features and better modeling the

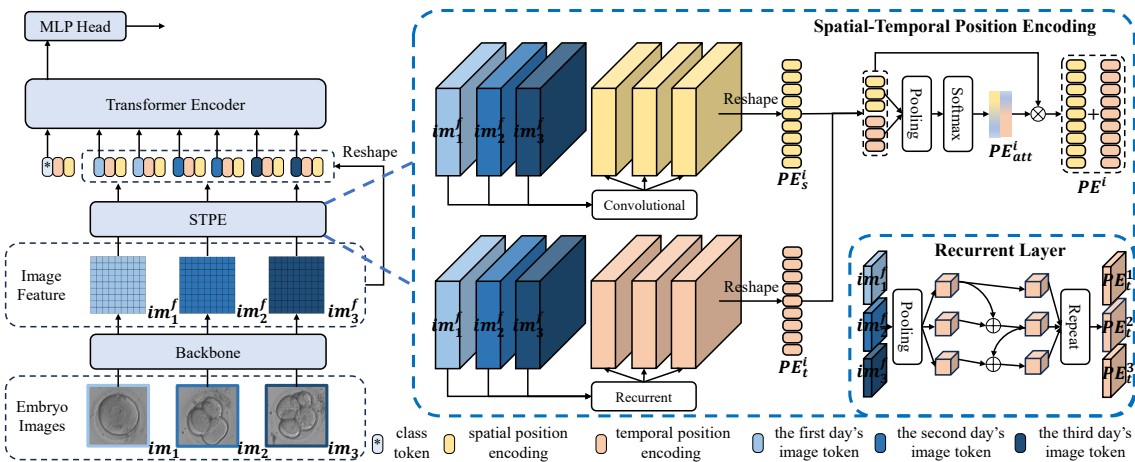

Figure 3: The details of temporal fusion network (image extractor).

complex interactions between modalities. So we propose the decoupling fusion module, a simple and effective multi-modal fusion module as showed in Fig. 2. The input information of the model are temporal grayscale embryo images and fertility table indicator. Embryo images are denoted as $\mathbf{im_i} \in \mathbb{R}^{1 \times \mathbf{H} \times \mathbf{W}}$, where $\mathbf{i} \in 1, 2, 3$ denote different days, and $\mathbf{H}$ and $\mathbf{W}$ denote the height and the width of an image, respectively. Table indicators are denoted as $\mathbf{ta} \in \mathbb{R}^{\mathbf{N}}$, where $\mathbf{N}$ denotes the number of indicators.

## 2.1. Image Extractor

To fuse the first three days embryo images for pregnancy prediction, we design a temporal image fusion network (image extractor). This network consists of three parts as shown in Fig. 3: a backbone aiming at extracting image features; a spatial-temporal position encoding (STPE) to obtain spatial information within a single image and temporal information among different images; and a Transformer (Dosovitskiy et al., 2021) that combine STPE to fuse temporal image features. Firstly, we use the backbone to extract embryo image features $\mathbf{im_i^f} \in \mathbb{R}^{\mathbf{C} \times \frac{\mathbf{H}}{\mathbf{S}} \times \frac{\mathbf{W}}{\mathbf{S}}}$ from the first three days, where $\mathbf{C}$ is the number of channels and $\mathbf{S}$ is the scaling factor. Then, we reshape the image features into tokens and add positional information to them. We encode image features as spatial position encoding $\mathbf{PE_s^i} \in \mathbb{R}^{\mathbf{C} \times \frac{\mathbf{H}}{\mathbf{S}} \times \frac{\mathbf{W}}{\mathbf{S}}}$ through a convolution layer: $\mathbf{PE_s^i} = \mathbf{Conv(im_i^f)}$. By using convolution operation to capture local spatial features, we can obtain spatial position information within a single image (Chu et al., 2021). Image features are encoded as temporal position encoding $\mathbf{PE_t^i} \in \mathbb{R}^{\mathbf{C} \times 1 \times 1}$ through a pooling layer and a series of recurrent layers: $\mathbf{PE_t^1} = \mathbf{Pooling(im_1^f)}, \mathbf{PE_t^2} = \mathbf{Pooling(im_2^f)} + \mathbf{PE_t^1}, \mathbf{PE_t^3} = \mathbf{Pooling(im_3^f)} + \mathbf{PE_t^2}$. By using recurrent operations to capture temporal dependencies, we can obtain temporal position information among different images (Hochreiter and Schmidhuber, 1997). To align image features, the $\mathbf{PE_t^i}$ needs to be replicated to get $\mathbf{PE_t^i} \in \mathbb{R}^{\mathbf{C} \times \frac{\mathbf{H}}{\mathbf{S}} \times \frac{\mathbf{W}}{\mathbf{S}}}$. The $\mathbf{PE_s^i}$ and $\mathbf{PE_t^i}$ are integrated through a position encoding attention:

$$\mathbf{PE_{att}^i} = \mathbf{Softmax(Pooling(PE_s^i) \| Pooling(PE_t^i))}, \tag{1}$$

where $\mathbf{PE^i_{att}} \in \mathbb{R}^{\frac{\mathbf{H \times W}}{\mathbf{S \times S}} \times \mathbf{2}}$. The final $\mathbf{PE^i}$ is as follows:

$$\mathbf{PE^i = PE^i_{att}[:, 0] * PE^i_s + PE^i_{att}[:, 1] * PE^i_t,} \tag{2}$$

where $\mathbf{PE^i} \in \mathbb{R}^{\frac{\mathbf{H \times W}}{\mathbf{S \times S}} \times \mathbf{C}}$. After obtaining the STPE, integrating the tokens obtained through image feature reshaping into the transformer encoder can effectively fuse temporal embryo images for pregnancy prediction.

## 2.2. Table Extractor

Inspired by TabTransformer (Huang et al., 2020) (table extractor), we extract table information of fertility table indicators by transformer (Vaswani et al., 2017). In order to adapt the table information to the transformer, we specify tabular embedding as a linear layer to upscale the table features $\mathbf{ta}$ to $\mathbf{ta^f} \in \mathbb{R}^{\mathbf{N \times 32}}$. Next, we construct a series of transformer layer with multi-head self-attention (MHSA) to extract table information:

$$\mathbf{Q = MLP(ta^f), K = MLP(ta^f), V = MLP(ta^f),}$$
$$\mathbf{\overline{ta^f} = MHSA(Q, K, V) + ta^f, \overline{\overline{ta^f}} = MLP(LN(ta^f)) + \overline{ta^f},} \tag{3}$$

here, $\mathbf{LN}$ means layer-norm and $\mathbf{MLP}$ is the linear layer.

## 2.3. Decoupling Fusion Module

We use an image feature extractor and a table feature extractor to extract temporal image features and table features. And we denote their outputs as $\mathbf{f_i}$ and $\mathbf{f_t}$, respectively. In order to fuse information from different modalities at a finer granularity, as shown in Fig. 2, we propose a decoupling fusion module that decouples the feature of different modalities into related (common) feature and unrelated (unique) feature. We extract related feature $\mathbf{f^c_i}$ and $\mathbf{f^c_t}$ between modalities through shared $\mathbf{E_c(f_m; \theta^c)}$, and extract unrelated feature $\mathbf{f^u_i}$ and $\mathbf{f^u_t}$ through $\mathbf{E_u(f_i; \theta^u_i)}$ and $\mathbf{E_u(f_t; \theta^u_t)}$. We can decouple the features of different modalities by using cross reconstruction method (Ji et al., 2021; Liu et al., 2023b). The cross reconstruction loss is as follows:

$$\mathcal{L_{recon}} = \sum_{m=0}^{M} ||\mathbf{f^m_i - D(f^c_t, f^u_i; \theta_i)}||_1 + \sum_{m=0}^{M} ||\mathbf{f^m_t - D(f^c_i, f^u_t; \theta_t)}||_1, \tag{4}$$

where $||.||_1$ is the L1-norm, $\mathbf{M}$ is the dimensionality of the feature, $D(f^c_t, f^u_i; \theta_i)$ and $D(f^c_i, f^u_t; \theta_t)$ are decoder. We obtain the final pregnancy prediction result by fusing the decoupled common and unique features: $\mathbf{y_b = Classifier(f_{\bar{i}}, f_{\bar{t}})}$, which is composed of three layers of MLP. Finally, we apply cross entropy loss to minimize the difference between the predicted results and the true labels. The classification loss is as follows:

$$\mathcal{L_{ce}} = -\frac{1}{\mathbf{B}} \sum_{b=0}^{B} [\mathbf{\overline{y}_b \log y_b + (1 - \overline{y}_b) \log(1 - y_b)}], \tag{5}$$

where $\mathbf{\overline{y}_b}$ represent true labels, $\mathbf{B}$ is the size of a batch. The overall loss function is as follows:

$$\mathcal{L = L_{ce}} + \lambda \mathcal{L_{recon}}. \tag{6}$$

where $\lambda$ is a hyperparameter in the loss function.

## 3. Experiments

### 3.1. Dataset

The first dataset used in the research is from Southern Medical University, with a total of 4046 valid embryo transfer cases. Each case includes both image data and tabular data. The image data are the first three days' microscopic images of embryo development. The tabular data include 22 parental fertility indicators. The label of each example is positive or negative, representing whether having fetal hearts successfully or not. We conduct a 5-fold cross validation on the dataset.

The second dataset comes from the Peking University International Competition on Ocular Disease Intelligent Recognition (ODIR) (ODIR, 2019), and the original task was to classify eye diseases through image uni-modality. In order to make the dataset applicable to multiple modalities, we extract a total of 3500 cases of image modality and table modality information for eye disease prediction. Among them, the image modality consists of a eye image, and the table modality consists of 8 indicators converted from keywords. We have already made this dataset accessible to the public as a new multi-modal dataset. We conduct a 4-fold cross validation on the dataset.

### 3.2. Evaluation Metric and Experimental Settings

In the experiments, we evaluate the performance with Accuracy, Area Under the ROC (AUC), and F1-score. AUC is a comprehensive metric to evaluate prediction accuracy. F1-score is an index taking into account the precision and recall of the model predictions.

We implement our method with PyTorch on a Nvidia GeForce RTX 2080ti graphics processing unit (GPU). In addition, since the image class tokens and the table class tokens need to share an encoder, we align them with a linear layer. The learning rate of the image extractor is 1e-6, while the learning rate of the table extractor model is 1e-4. The learning rate of the DeFusion model is 1e-5. The above models all use the Adam (Kingma and Ba, 2014) optimizer. As in Section 3, $H = 224$, $W = 224$, $N = 22$, and $\lambda = 1$.

### 3.3. Baseline Methods

We compare our method with baseline methods as shown in Table 1, our model achieves superior performance in all the evaluation metrics. Firstly, for table modality, we compare a **SVM** (Dehghan et al., 2024) and a **Adaboost** (Dehghan et al., 2024) as uni-modal models based on parental tabular fertility indicators. We also compare **TabNet** (Arik and Pfister, 2021), which is a neural network model specifically designed for tabular classification tasks. The TabTransformer method we use has the best performance. Secondly, for image modality, by comparing temporal image fusion models based on Add, LSTM (Hochreiter and Schmidhuber, 1997), and Transformer with different positional encodings (sin-cos and learnable) (Vaswani et al., 2017), our STPE achieve optimal performance in the transformer-based temporal image fusion strategy. Thirdly, for image and table modalities, we compare the recent multi-modal baseline methods have been introduced in Appendix A and B. Our method, as a new category fusion approach, achieves optimal results.

Table 1: Comparative and ablation experiment results for pregnancy prediction.

| Modality | Method | AUC | F1 | Accuracy |
|---|---|---|---|---|
| Table | MLP | 0.684(0.006) | 0.649(0.012) | 0.641(0.013) |
| | SVM | 0.690(0.006) | 0.643(0.011) | 0.634(0.011) |
| | Adaboost | 0.708(0.009) | 0.631(0.016) | 0.640(0.015) |
| | TabNet | 0.702(0.014) | 0.643(0.009) | 0.634(0.010) |
| | **Ours(TabTransformer)** | **0.713(0.012)** | **0.661(0.011)** | **0.653(0.006)** |
| Image | ResNet+Add | 0.554(0.008) | 0.584(0.012) | 0.592(0.014) |
| | ResNet+LSTM | 0.572(0.014) | 0.591(0.018) | 0.600(0.026) |
| | ResNet+Learnable | 0.589(0.012) | 0.597(0.013) | 0.598(0.020) |
| | ResNet+SinCos | 0.602(0.011) | 0.607(0.014) | 0.612(0.021) |
| | **Ours(ResNet+STPE)** | **0.617(0.012)** | **0.621(0.022)** | **0.631(0.016)** |
| | w/o SPE | 0.614(0.013) | 0.611(0.013) | 0.627(0.015) |
| | w/o TPE | 0.596(0.011) | 0.607(0.008) | 0.613(0.007) |
| | w/o PEAttention | 0.600(0.009) | 0.605(0.007) | 0.613(0.020) |
| | w/o STPE | 0.584(0.016) | 0.603(0.015) | 0.619(0.021) |
| | with ViT | 0.604(0.009) | 0.609(0.013) | 0.613(0.016) |
| Image and Table | MMBE | 0.723(0.004) | 0.682(0.021) | 0.681(0.011) |
| | MOAB | 0.719(0.003) | 0.681(0.009) | 0.674(0.011) |
| | SFusion | 0.718(0.005) | 0.667(0.013) | 0.658(0.013) |
| | ConGraph | 0.718(0.011) | 0.649(0.009) | 0.640(0.009) |
| | HMCAT | 0.723(0.011) | 0.655(0.014) | 0.646(0.012) |
| | **Ours(DeFusion)** | **0.746(0.003)** | **0.689(0.017)** | **0.691(0.010)** |
| | w/o Decoupling Module | 0.715(0.007) | 0.689(0.013) | 0.681(0.014) |
| | with TabNet | 0.735(0.010) | 0.681(0.011) | 0.683(0.013) |

## 3.4. Ablation Study

We conduct ablation experiments to evaluate the contribution of each module in our model in Table 1. We first ablate the Spatial-Temporal Position Encoding (STPE) including Spatial Position Encoding (SPE), Temporal Position Encoding (TPE) Position Encoding Attention (PEAttention) titled as **w/o SPE**, **w/o TPE**, **w/o PEAttention** and **w/o STPE**, respectively. In addition, by comparing the performance of ResNet (He et al., 2016) and ViT (**with ViT**) (Dosovitskiy et al., 2021), we choose ResNet as the backbone of the image. Similarly, we evaluate the contribution of the Decoupling Module through **w/o Decoupling Module**. In addition, by comparing TabNet (Arik and Pfister, 2021) (**with TabNet**) and TabTransformer, we choose TabTransformer as the table extractor.

## 3.5. Generalization

Although the proposed DeFusion model is designed for pregnancy prediction, the principles behind it are universal and can be transferred to other multi-modal medical image analysis tasks. We extend DeFusion for multi-modal eye disease prediction on the ODIR dataset. The final prediction results are shown in the Table 2. Although our model doesn't achieve the highest accuracy, it performs best in the AUC metric, indicating that our model performs better in terms of overall performance.

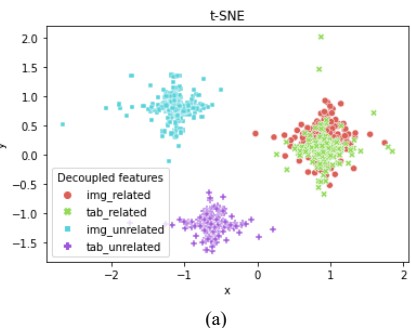
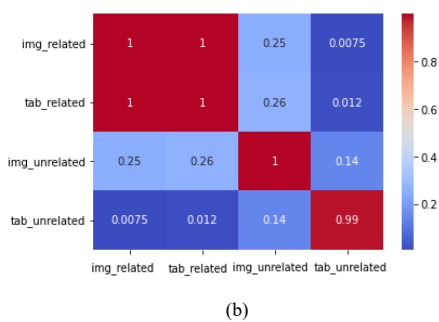

(a)                                                (b)

Figure 4: (a) Decoupled features of the test dataset visualized in a t-SNE space. (b) PCC matrix of the decoupled features.

Table 2: Comparative and ablation experiment results in ODIR dataset.

| Method | AUC | F1 | Accuracy |
|---|---|---|---|
| MMBE | 0.836(0.007) | 0.765(0.009) | 0.766(0.007) |
| MOAB | 0.827(0.004) | 0.756(0.005) | 0.751(0.004) |
| SFusion | 0.791(0.030) | 0.755(0.013) | 0.755(0.010) |
| ConGraph | 0.836(0.009) | 0.756(0.008) | 0.751(0.008) |
| HMCAT | 0.835(0.006) | 0.773(0.017) | 0.772(0.010) |
| **Ours(DeFusion)** | **0.842(0.004)** | **0.772(0.009)** | **0.770(0.009)** |
| w/o Decoupling | 0.825(0.003) | 0.763(0.014) | 0.759(0.014) |

### 3.6. Visualization

As Fig. 4 shows, we output t-SNE (van der Maaten and Hinton, 2008) results and average Pearson correlation coefficient (PCC) matrix (Sverko et al., 2022) of $\mathbf{f_i^c}$(img_related), $\mathbf{f_t^c}$(tab_related), $\mathbf{f_i^u}$(img_unrelated) and $\mathbf{f_t^u}$(tab_unrelated) from the decoupling test set. The PCC is between 0 and 1, with a larger value indicating greater relevance. The overlap between the points of $\mathbf{f_i^c}$ and $\mathbf{f_t^c}$ after t-SNE dimensionality reduction and the high PCC value between $\mathbf{f_i^c}$ and $\mathbf{f_t^c}$ indicate that the model successfully capture relevant and overlapping information between the two modalities. On the contrary, $\mathbf{f_i^u}$ and $\mathbf{f_t^u}$ are well separated, indicating it capture the information that is independent and complementary between the two modalities. These prove the effectiveness of the decoupling module.

### 4. Conclusion

This paper proposes a Decoupling Fusion Network called DeFusion to integrate the multi-modal information of temporal embryo images and parental fertility table indicators for IVF-ET pregnancy prediction. The superior performance suggest that our model can provide valuable assistance for the selection of embryos for transplantation. And the effectiveness of the decoupling fusion module has been demonstrated through visualization and generalization experiments. In the future, we will optimize the decoupling module and expand it to more datasets.

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

## Appendix A. Related Work

Fusion of heterogeneous information from multi-modal data can effectively enhance model performance, which is a key project in the medical field of multi-modal learning (Cui et al., 2023). Decision-level fusion and feature-level fusion are two main strategies for multi-modal fusion. The decision-level fusion employs averaged, weighted voting or majority voting (Holste et al., 2021) to integrate the outputs of uni-modal models, so as to make the final multi-modal output.

Although decision-level fusion is simple to implement, it cannot capture the interactions between hidden features from different modalities. The feature-level fusion fuses the heterogeneous multi-modal data by projecting extracted features into a compact and information-rich multi-modal hidden representation space. Feature-level fusion mainly includes simple-operation based, tensor-based, transformer-based and graph-based methods. The simple-operation method performs concatenation, element addition and element multiplication operations. (Zhou et al., 2021) use two branch encoders to extract image and non-image information, and fuse the extracted information on key point through simple operation for COVID-19 patient severity prediction. The tensor-based method performs outer product between multi-modal feature vectors to form higher-order co-occurrence matrices, which provide more informative information than these features alone. (Zolotarev et al., 2024)(**MOAB**) use a deep orthogonal fusion model to predict the atrial fibrillation from different multi-modal data. The attention mechanism in the transformer-based method has the ability to aggregate features in different feature spaces, making it very suitable for multi-modal alignment and fusion. (Liu et al., 2023c)(**SFusion**) apply transformer to fuse different modalities of brain imaging for tumor segmentation; (Li et al., 2023b)(**HMCAT**) use a model based on the cross attention transformer that integrates pathological and radiological images for cancer prediction. Graph-based modeling and inference can provide a deeper understanding of disease information by discovering complex relationships between hidden disease tissue regions. (Ding et al., 2024)(**ConGraph**) transform images and non-images into graph nodes based on fully connected graph attention network, and fuse information among nodes to predict Pakinson's disease.

## Appendix B. Baseline Methods

### B.1. Baseline methods with image modality

**ReNet+Add** applys ResNet to extract image information $F_i \in \mathbb{R}^{512}$ from the first three days of embryonic development, where $i \in 1, 2, 3$. Then, $F_1$, $F_2$ and $F_3$ are fused through an addition operator. Finally, we use a classifier consisting of three non-linear layers for pregnancy prediction. **ReNet+LSTM** replaces the addition operator with LSTM on the basis of **ReNet+Add**. **No Position**, **Learnable** and **Sin-Cos** are the results of replacing our proposed spatial-temporal position encoding with different position encoding.

### B.2. Baseline methods with both image modality and table modality

As shown in Fig. A.1, we compare different multi-modal fusion methods. To ensure fairness in comparison, all methods use the same backbone. In addition, according to the structural

characteristics of different models, TransformerFusion (**SFusion** (Liu et al., 2023c), **HM-CAT** (Li et al., 2023b)) and GraphFusion (**ConGraph** (Ding et al., 2024)) use all image and table tokens, while AddFusion (**MMBE** (Liu et al., 2022)), TensorFusion (**MOAB** (Zolotarev et al., 2024)), and our DecouplingFusion (**DeFusion**) only use image class token and table class token.

As shown in Fig. A.1 (a) and (b), **AddFusion** and **TensorFusion** directly fuse the class tokens of the two modalities using an addition operation and an outer product operation, respectively. As shown in Fig. A.1 (c) and (d), **TransformerFusion** and **GraphFusion** use the Graph Attention Network (Velickovic et al., 2017) and the Transformer (Dosovitskiy et al., 2021) as the fusion network to fuse all tokens of the two modalities, respectively. Our method is an innovative multi-modal fusion category.

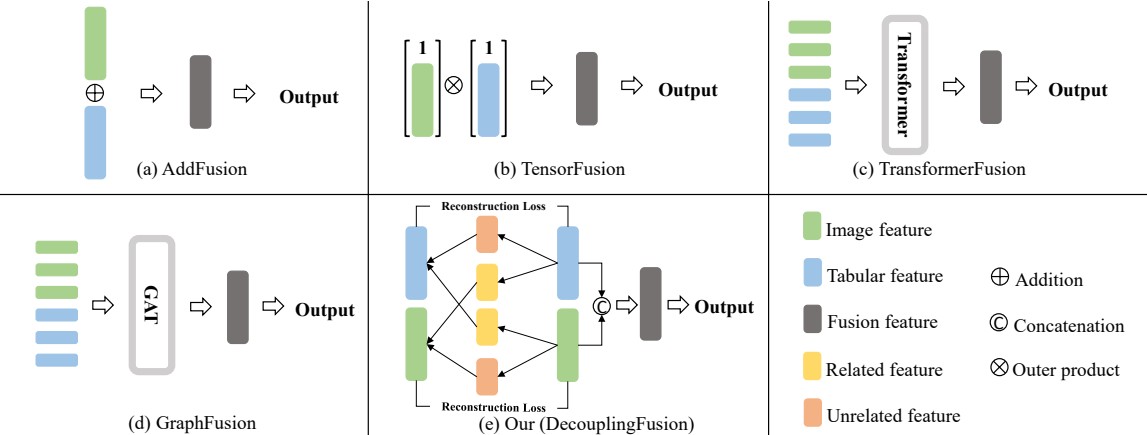

Figure A.1: Comparison of different multi-modal fusion frameworks.

## Appendix C. Dataset Samples

In the Fig.A.2 and Table A.1, we present some examples of image modality and table modality used for pregnancy prediction, respectively. When processing image data, we use the following data enhancement during training and testing: resize to 256 pixels and then center cropping to 224 pixels. Finally, normalization with a mean of 0.566 and a variance of 0.063 is used, and the mean and variance were obtained by statistics of the whole data set. When processing tabular data, we use the average of features to replace missing features. Then we input the features into the neural network after min-max normalization.

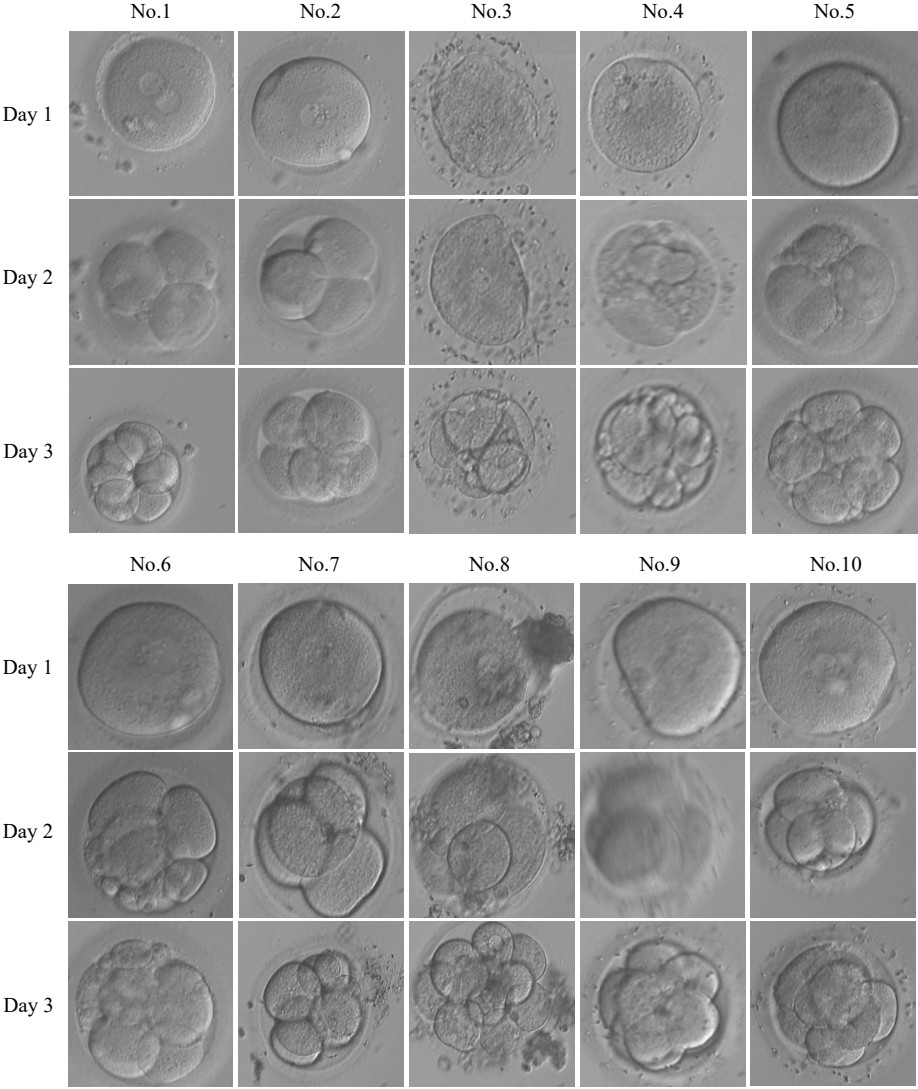

Figure A.2: Ten examples of temporal embryo images. Each column represents an example, containing images of the first, second, and third day of embryonic development.

Table A.1: Ten examples of parental fertility indicators. BMI represents Body Mass Index, AFC represents Antral Follicle Counting, HCG represents Human Chorionic Gonadotropin, E2 represents Estradiol, FSH represents Follicle-Stimulating Hormone, BT represents Before Semen Treatment, AT represents After Semen Treatment and - represents the missing value.

| Number | 1 | 2 | 3 | 4 | 5 | 6 | 7 | 8 | 9 | 10 |
|---|---|---|---|---|---|---|---|---|---|---|
| Female age | 39 | 25 | 44 | 34 | 31 | 36 | 33 | 29 | 28 | 29 |
| Man age | 46 | 24 | 45 | 36 | 35 | 38 | 38 | 32 | 35 | 31 |
| Female BMI | 22.8 | 18.7 | 22.1 | 21.5 | 18.5 | 23.9 | 23.4 | 18 | 25 | 20.4 |
| AFC | 10 | - | - | 8 | 18 | 9 | 12 | 16 | 10 | 8 |
| Number of obtained oocytes | 5 | 22 | 3 | 10 | 17 | 14 | 5 | 11 | 7 | 7 |
| Number of mature oocytes | 4 | 14 | 3 | 10 | 15 | 10 | 5 | 8 | 7 | 7 |
| Available embryos | 3 | 2 | 3 | 7 | 7 | 3 | 4 | 3 | 5 | 7 |
| High-quality embryos | 2 | 1 | 1 | 7 | 5 | 1 | 2 | 2 | 4 | 4 |
| HCG Day E2 | 1037 | 2271 | 1782 | 1483 | 3118 | 3693 | 731.9 | 1891 | 1309 | 3767 |
| HCG intimal thickness | 7.5 | 15.5 | 8 | 9.3 | 9.4 | 11.7 | 9.8 | 15 | 17.5 | 12.5 |
| FSH | 8 | - | - | 7.81 | 7.22 | 4.07 | 4.61 | 6.53 | 7.3 | - |
| Infertility years | 2 | - | - | 9 | - | 7 | 10 | 1 | 3 | 4 |
| Volume BT | 1.4 | 4.1 | 4 | 2.6 | 2 | 1.5 | 1.5 | 1 | 2.5 | 0.3 |
| Concentration BT | 6 | 2 | 25 | 20 | 0.01 | 3 | 4 | 10 | 18 | 40 |
| Non forward movement BT | 5 | 10 | 5 | 15 | 0.5 | 5 | 3 | 10 | 10 | 5 |
| Inactivity BT | 90 | 80 | 85 | 60 | 0.5 | 90 | 95 | 85 | 80 | 75 |
| Forward movement BT | 5 | 10 | 10 | 25 | 0.5 | 5 | 2 | 5 | 10 | 10 |
| Volume AT | 0.1 | 0.1 | 0.6 | 0.2 | 0.2 | 0.15 | 0.2 | 0.15 | 0.3 | 0.3 |
| Concentration AT | 1 | 2 | 3 | 8 | 0.01 | 1 | 1 | 1 | 1 | 3 |
| Non forward movement AT | 20 | 10 | 5 | 5 | 0.5 | 10 | 20 | 20 | 20 | 10 |
| Inactivity AT | 30 | 25 | 5 | 5 | 0.5 | 10 | 10 | 40 | 10 | 5 |
| Forward movement AT | 50 | 65 | 90 | 90 | 0.5 | 80 | 70 | 30 | 60 | 85 |

## Appendix D. Supplementary Experiments

### D.1. Table Modality Experiments

As shown in Fig.A.3, it is a detail of the table extractor (TabTransformer).

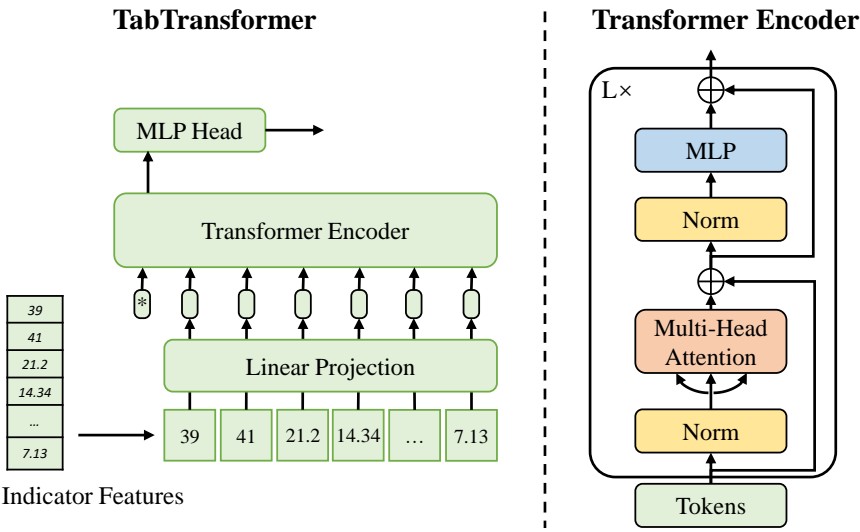

Figure A.3: The details of TabTransformer (table extractor).

### D.2. Image Modality Experiments

Table A.2 shows the performance with image modality only. We use images of embryo development on the first, second, and third day for pregnancy prediction. And the backbone is ResNet. The results indicate that as the embryo continues to develop, it provides greater assistance in predicting pregnancy.

Table A.2: Comparative experiment results for pregnancy prediction with image modality.

| Modality | Method | AUC | F1 | Accuracy |
|---|---|---|---|---|
| Image | ResNet(First Day) | 0.476(0.004) | 0.515(0.009) | 0.502(0.009) |
| | ResNet(Second Day) | 0.565(0.007) | 0.572(0.013) | 0.560(0.013) |
| | ResNet(Third Day) | 0.593(0.007) | 0.598(0.012) | 0.595(0.010) |
| | **Ours(ResNet+STPE)(Three Days)** | **0.617(0.012)** | **0.621(0.022)** | **0.631(0.016)** |

### D.3. Computational Complexity

As shown in the Table A.3, we compare the computational complexity of multi-modal fusion methods. Although our model is not optimal in terms of computational complexity, compared to other methods, our approach still achieves competitive results in terms of computational complexity while achieving optimal accuracy.

Table A.3: Comparison of computational complexity of different multi-modal fusion methods. GFLOPs, the smaller the index, the better. Training time (in seconds) of a single epoch on 2080Ti GPU with 12G memory, the smaller the index, the better. Frames per second (FPS) on the i7-6850K@3.60GHz CPU, the larger the better.

| Method | MMBE | MOAB | SFusion | ConGraph | HMCAT | Ours(DeDusion) |
|---|---|---|---|---|---|---|
| GFLOPs($\downarrow$) | 12.66 | 13.30 | 13.66 | 14.01 | 16.19 | 12.67 |
| Training Time($\downarrow$) | 50 | 91 | 65 | 68 | 79 | 64 |
| Inference FPS($\uparrow$) | 5.05 | 2.69 | 3.34 | 3.54 | 2.79 | 3.51 |

## D.4. Uni-modal Experiment on The ODIR

As shown in the Table A.4, we compare the uni-modal method on the ODIR dataset for eye disease prediction.

Table A.4: Uni-modal comparison on the ODIR dataset for eye disease prediction.

| Modality | Method | AUC | F1 | Accuracy |
|---|---|---|---|---|
| Image | ResNet | 0.714(0.016) | 0.677(0.015) | 0.674(0.015) |
| | ViT | 0.736(0.006) | 0.685(0.013) | 0.680(0.014) |
| Table | SVM | 0.802(0.003) | 0.722(0.001) | 0.714(0.007) |
| | TabNet | 0.793(0.024) | 0.708(0.023) | 0.701(0.015) |
| | MLP | 0.793(0.001) | 0.731(0.007) | 0.727(0.003) |

## D.5. Generalization of Other Dataset

We collect a dataset of 218 cases from Guangzhou Women and Children's Medical Center as an additional test set to test our DeFusion model, with 56 pregnant cases and 162 non-pregnant cases in this dataset. As shown in the Table A.5, our model has certain generalization ability without fine tuning.

Table A.5: Generalization experiments of the model on other hospital datasets.

| Modality | Method | AUC | F1 | Accuracy |
|---|---|---|---|---|
| Image+Table | Ours(DeFusion) | 0.616 | 0.660 | 0.642 |

## Appendix E. Interpretability

In order to analyze the reasons for the success of the model, we conduct interpretability analysis on the model, mainly manifest in two aspects. Firstly, as shown in Fig. A.4, we use SHAP (Lundberg and Lee, 2017) to output the importance ranking of clinical indicators in our model, which focuses more on features such as female age, high-quality embryos and so on. Among them, A.4(a) ranks the importance of table features. A.4(b) is a beeswarm, which depicts the SHAP values of each sample under different features.

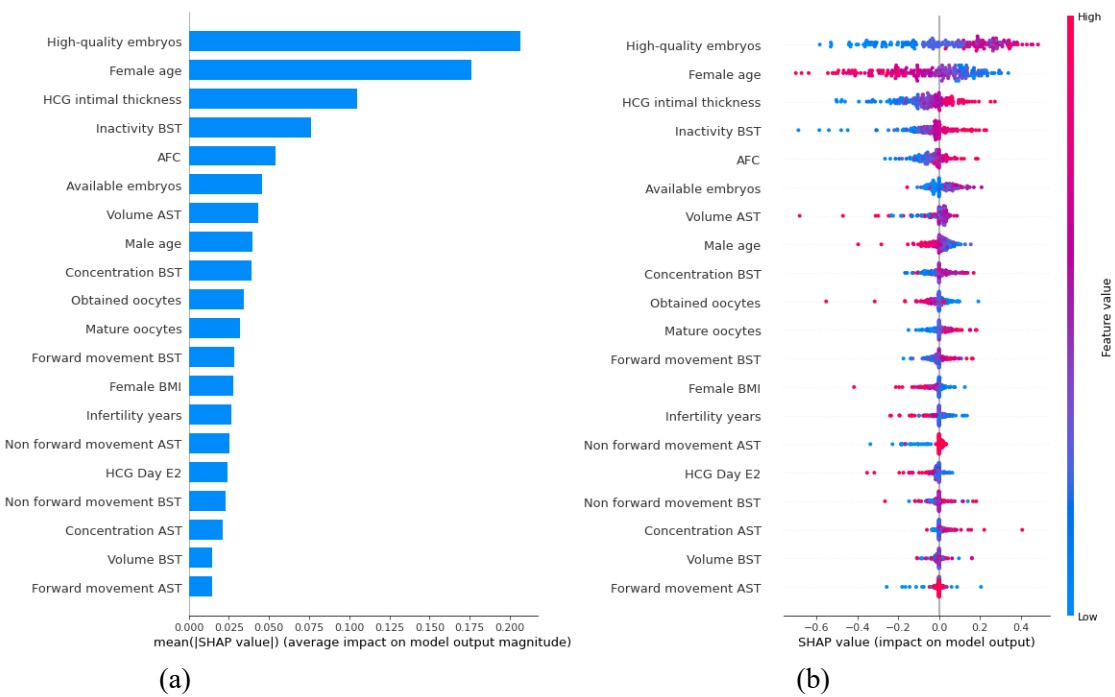

Figure A.4: SHAP interpretability of clinical tabular indicators.

Secondly, we use the Grad-Cam (Selvaraju et al., 2017) to visualize the class activation maps of the first three days of embryonic development images in Fig. A.5. We can see the areas that our model focuses on are the edges of embryonic cells. These interpretable results are consistent with the experience of obstetricians and gynecologists.

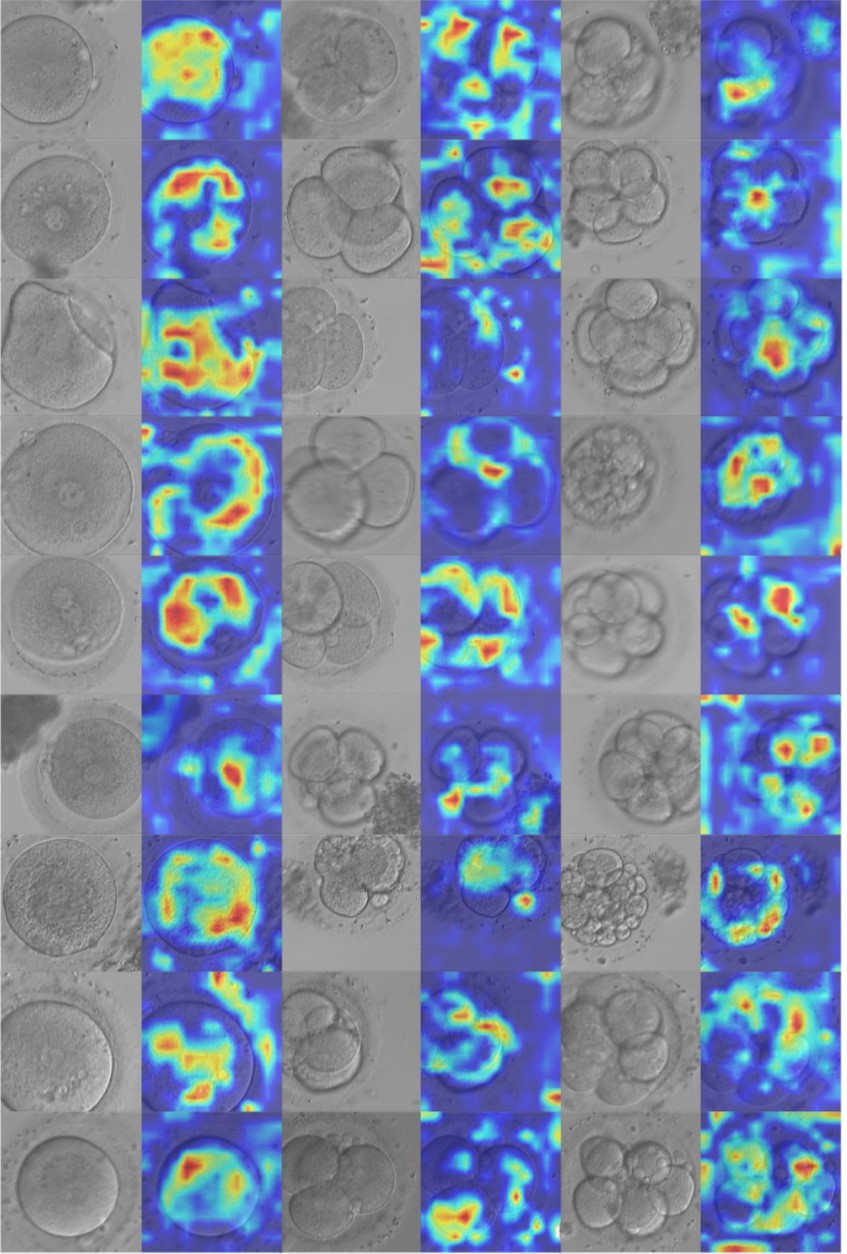

Figure A.5: Grad-Cam visualization of embryonic images, with each row representing a three-day image.

## Appendix F. Limitation

Although our study is superior to other methods, there are limitations to our method and data set. For our method, our premise is that there is correlation between the multi-modal data. If there is no correlation between the multi-modal data, our method may not work well. In addition, when decoupling unique features and common features, we only use a simple cross-reconstruction loss constraint, which is very weak. Although the decoupling visualization by t-SNE and Pearson correlation coefficient proves the effectiveness of the decoupling method, adding stronger loss function constraints to the decoupling process may make the decoupling process smoother. Finally, because we add the decoupling module to the multi-modal fusion process, our computational complexity is higher than the simple ADD fusion, which is not conducive to our deployment of the model to the end-to-end device.

For our dataset, we only collected 4046 cases of data, which is not enough for deep learning. And, there are some missing values in our clinical indicator data, which is unfavorable to the prediction of results. In addition, our image data is three images taken every other day using a normal microscope, and many studies now use time-lapse microscopes, which can acquire images at the hour or even minute level, so our image data lacks a lot of temporal information compared to other studies.

