# OpenReview forum: "DeFusion: An Effective Decoupling Fusion Network for Multi-Modal Pregnancy Prediction"
_MIDL.io/2025/Conference — MIDL 2025 Poster_

### Official Review · Reviewer_xL45 · 2025-02-18

**Confidence:** 5
**Preliminary Rating:** 4
**Recommendation:** Oral

**Summary:**

The paper "DeFusion: An Effective Decoupling Fusion Network for Multi-Modal Pregnancy Prediction" introduces a novel multi-modal deep learning model to improve pregnancy prediction in IVF-ET (in vitro fertilization-embryo transfer). The DeFusion network effectively integrates temporal embryo images (from the first three days of development) with parental fertility indicators using a decoupling fusion module, which distinguishes common vs. unique modality-specific features. The study evaluates DeFusion on a dataset of 4046 IVF cases from Southern Medical University, using 5-fold cross-validation, achieving AUC = 0.746, F1 = 0.689, and accuracy = 0.691, outperforming existing models like MMBE and MOAB. Additionally, the model is tested on the ODIR eye disease dataset to assess its generalization ability, achieving AUC = 0.842. The study’s significance lies in its ability to enhance clinical decision-making, improve embryo selection accuracy, and provide a transparent, explainable AI model through SHAP and Grad-CAM visualizations. By making its code and dataset publicly available, this research contributes to advancing AI-driven reproductive medicine.

**Strengths:**

One of its key strengths is the introduction of a Decoupling Fusion Module (DFM), which uniquely separates modality-common and modality-unique features, enhancing the interpretability and effectiveness of multi-modal fusion. Unlike prior works that rely on single-day embryo images or tabular fertility indicators alone, this paper integrates three days of temporal embryo images with 22 parental fertility indicators, making it clinically more relevant and robust for real-world IVF scenarios.

The methodology is well-structured, with a strong theoretical foundation and rigorous experimental validation. The authors perform 5-fold cross-validation on a dataset of 4046 IVF cases, ensuring reliable results. The model’s performance is evaluated using AUC, F1-score, and accuracy, showing an AUC of 0.746, which outperforms existing methods like MMBE, MOAB, and SFusion. Furthermore, the generalization study on the ODIR eye disease dataset demonstrates that the proposed framework can be adapted to different medical applications, making it valuable to the broader medical AI community.

Another major strength is the explainability of the model. The paper uses SHAP to analyze the importance of fertility indicators and Grad-CAM to visualize embryo feature activations, ensuring that the model’s predictions are interpretable for clinicians. The inclusion of publicly available code further enhances its impact by enabling reproducibility and fostering future research.

The language, structure, and scientific presentation of the paper are of high quality, with clear explanations, well-organized sections, and properly formatted equations, figures, and references. The work also adequately reviews prior research, comparing its method against multiple

**Weaknesses:**

One potential weakness is the limited external validation beyond the datasets used. The primary dataset consists of 4046 IVF cases from a single medical institution, which may introduce institutional bias. Although the model is tested on the ODIR dataset for generalization, this dataset pertains to eye disease classification, which does not directly validate the model’s robustness in other reproductive health settings. A more comprehensive external validation on multi-center IVF datasets would enhance the reliability of the findings.

Another concern is the computational complexity of the proposed approach. While DeFusion achieves superior performance, it requires a multi-modal deep learning architecture with a decoupling fusion module, which increases training time and demands high computational resources. The paper provides GFLOPs comparisons, showing that DeFusion is competitive, but further analysis on real-world feasibility (e.g., model deployment in resource-constrained clinical settings) would be beneficial.

Additionally, the dataset availability poses a limitation. While the authors provide code on GitHub, the IVF dataset is not fully publicly available, making it challenging for other researchers to replicate and extend the work. Publicly accessible multi-modal IVF datasets would significantly increase the impact and adoption of the proposed method.

Lastly, while the paper reviews prior work extensively, more discussion on potential failure cases and limitations of decoupling fusion would strengthen the study. Understanding scenarios where DeFusion may struggle, such as cases with limited fertility indicator data or missing embryo images, would provide deeper insights into its applicability in clinical practice.

**Detailed Comments:**

1-The paper demonstrates strong generalization by testing on the ODIR dataset (eye disease prediction), but this dataset is unrelated to IVF-ET.
Consider evaluating on a multi-center IVF dataset or using an external held-out test set to further validate clinical applicability.

2- The DeFusion model introduces a decoupling fusion module, which improves performance but adds computational overhead.
Consider discussing inference speed and resource requirements for real-world deployment in clinical settings.
A comparison of training time and inference latency with other fusion models would be helpful.

**Justification Of The Preliminary Rating:**

he paper presents a strong and novel approach to multi-modal fusion for medical prediction, specifically in the challenging task of IVF-ET pregnancy prediction. The Decoupling Fusion Module (DFM) effectively separates modality-common and modality-unique features, improving fusion granularity and interpretability, which is an important contribution to the field of medical AI fusion techniques. The use of distinct data modalities (temporal embryo images + parental fertility indicators) and the generalization study using the ODIR dataset demonstrate the potential applicability of the model beyond IVF-ET. Additionally, the ablation studies provide strong evidence of the model’s effectiveness in multi-modal fusion scenarios.

However, while the 5-fold cross-validation (IVF-ET) and 4-fold cross-validation (ODIR) demonstrate strong performance, the lack of a completely separate test set raises concerns about true generalization. It is highly recommended that the authors evaluate the model on an independent test dataset beyond cross-validation to confirm robustness. Additionally, while using ODIR as a generalization test is an interesting approach, its relevance to IVF prediction could be better justified.

Thus, while the paper excels in innovation, methodology, and performance, the absence of a fully independent test set and the need for further justification of dataset choices prevents a higher rating at this stage. Addressing these concerns in the rebuttal phase would significantly strengthen the paper's contribution and impact.

**Questions To Address In The Rebuttal:**

1-Do the authors have access to another independent IVF dataset (from a different hospital or country)? If not, can they hold out a portion of the current dataset as an independent test set (rather than just cross-validation)?
2-Could they offer detailed preprocessing steps to allow others to replicate their approach with different data (Omics, EHR, Images,..)?
3-Can they discuss potential limitations due to dataset specificity?

**Special Issue:**

No

---

> ### Author Response · Authors · 2025-03-07
>
> We sincerely thank the reviewer for the thoughtful feedback. Below, we will answer the general concerns.
> 1. We collect a dataset of 218 cases from Guangzhou Women and Children's Medical Center as an additional test set to test our DeFusion model, with 56 pregnant cases and 162 non-pregnant cases in this dataset. Without doing any fine tuning, AUC=0.616, ACC=0.642, and F1-score=0.660.
> 2. For the study of computational complexity, besides dealing with GFLOPs in Table A.2, we also added the training time (in seconds) of a single epoch on 2080Ti GPU with 12G memory, the smaller the index, the better. And Frames Per Second (FPS) on the i7-6850K@3.60GHz CPU, the larger the better. The model, training time(↓), and test FPS(↑) results are as follows:
>
> MMBE: 50, 5.05;
>
> MOAB: 91, 2.69;
>
> SFusion: 65, 3.34;
>
> ConGraph: 68, 3.54;
>
> HMCAT: 79, 2.79;
>
> Ours(DeDusion): 64, 3.51.
>
> Our method is not optimal in computational complexity, which needs to be improved.
> 3. When processing image data, we use the following data enhancement during training and testing: resize to 256 pixels and then center cropping to 224 pixels. Finally, normalization with a mean of 0.566 and a variance of 0.063 is used, and the mean and variance were obtained by statistics of the whole data set. When processing tabular data, we use the average of features to replace missing features. Then we input the features into the neural network after min-max normalization.
> 4. Although our study is superior to other methods, there are limitations to our method and data set. For our method, our premise is that there is correlation between the multimodal data. If there is no correlation between the multimodal data, our method may not work well. In addition, when decoupling unique features and common features, we only use a simple cross-reconstruction loss constraint, which is very weak. Although the decoupling visualization by t-SNE and Pearson correlation coefficient proves the effectiveness of the decoupling method, adding stronger loss function constraints to the decoupling process may make the decoupling process smoother. Finally, because we add the decoupling module to the multimodal fusion process, our computational complexity is higher than the simple ADD fusion, which is not conducive to our deployment of the model to the end-to-end device.
> For our dataset, we only collected 4046 cases of data, which is not enough for deep learning. And there are some missing values in our clinical indicator data, which is unfavorable to the prediction of results. In addition, our image data is three images taken every other day using a normal microscope, and many studies now use time-lapse microscopes, which can acquire images at the hour or even minute level, so our image data lacks a lot of temporal information compared to other studies.
>
> We will supplement the above details in the official manuscript.

---

> > ### Comment · Area_Chair_ew4h · 2025-03-11
> >
> > Dear reviewer,
> >
> > The authors have now submitted a rebuttal to your concerns, could you please comment on whether they addressed your questions?

---

> > > ### Comment · Reviewer_xL45 · 2025-03-14
> > >
> > > Thank you for addressing my questions and considering my concerns in your rebuttal. I have no further inquiries and will maintain my initial rating

---

### Official Review · Reviewer_CUmP · 2025-02-19

**Confidence:** 4
**Preliminary Rating:** 4
**Recommendation:** Oral, Poster
**Final Rating:** 4

**Summary:**

This paper introduces a decoupling fusion network (DeFusion) to combine embryo images from days 1–3 of development with parental fertility indicators for in vitro fertilization (IVF) pregnancy prediction. It extracts common and unique features from these two modalities through dedicated encoders and enforces consistency with a cross-reconstruction loss. The approach achieves promising performance compared to both uni-modal and other multimodal baselines, and it also demonstrates encouraging results on an eye disease dataset to highlight generalizability. The study offers interesting insights into how early-stage imaging can be integrated with tabular data but would benefit from further experiments on publicly available IVF datasets and more clarity on some of the propsed concepts.

**Strengths:**

The paper addresses an important clinical scenario, proposing a method that fuses early-stage embryo images (days 1–3) and parental fertility indicators for direct pregnancy prediction. Its decoupling concept, separating each modality’s features into shared and unrelated parts, is well structured and potentially valuable for modeling complex multimodal data. The clarity in comparing different baselines—both uni-modal and multimodal—enables readers to appreciate the performance gains. Moreover, the paper includes interpretability analyses (e.g., Grad-CAM visualizations, SHAP for tabular features), aiding transparency into which features the model relies upon. The extension to a second dataset suggests that the framework can be adopted in other contexts, making it a potentially useful contribution to the broader field of medical AI. The availability of the code is also a plus.

**Weaknesses:**

Some points need further clarification. The terminology “order” versus “disorder” of decoupled features is not rigorously explained or quantified (e.g., using entropy or other measures), which weakens readers’ understanding of how the method captures shared vs. unique aspects. There are also places where abbreviations or variable names appear without clear definitions (e.g., E for encoder and D for decoder, or the repeated use of K for both “Keys” and dimensional indices). Comparing day 3 and day 5 embryos directly might better validate the authors’ claims about optimal time points. In addition, the paper occasionally uses phrases like “superior performance” where improvements are marginal; it is advisable to contextualize them. Finally, computational complexity, and a concrete discussion of the method’s limitations receive little attention and could be expanded.

**Detailed Comments:**

- Consider a more detailed explanation of the “order” vs. “disorder” concept: if you intend to convey a shift from entangled features to disentangled ones, clarify whether any measure (e.g., feature entropy, correlation, mutual information, or cluster separability) is used.
- Publicly available IVF datasets (for instance, https://doi.org/10.1038/s41597-023-02182-3) could bolster reproducibility and comparison with existing methods.
- When mentioning methods or acronyms, define them in full before using abbreviations.
- Including day 5 images would strengthen the claim about why day 3 images are the chosen focus and allow direct exploration of the optimal timing. Of course I understand that this will not be possible in the rebuttal phase, but would appreciate if you could consider exploring this in the future.
- Provide more details on the computational cost (e.g., GFLOPs, training time) across baselines.
- Why did you deliberately focus on representing patient data as tabular data? Patient data per se can also be seen as a simple vector of data points without the need for a large transformer. How did you come up with the term table modality?
- If simpler embeddings for tabular data (MLP or linear transformations) were tested and found suboptimal, mention and discuss these results to justify using a full transformer-based extractor.
- Consider investigating more advanced LSTM variants (e.g., xLSTM) to confirm that the temporal part of the model is well optimized. Similar to the comment regarding day 5 images, a future consideration of this would be interesting.
- I assume E denotes an encoder parameterized with θ, similar D denotes a decoder right? I’d appreciate declaring variables before use. Furthermore, the use of K for the transformer Keys as well as for the upper limit in equation 4 can be confusing.
- I’d appreciate more in depth explanations in the figure captions to ease the reader experience.
- A thorough proofread is advisable to correct minor grammatical slips (e.g., “predicting the outcome of a blastocyst”).

**Justification Of The Final Rating:**

Thank you for answering my questions and considering my concerns in your rebuttal. I have no further questions and will remain with my initial rating. The paper addresses a clinically valuable problem with a valid multi-modal fusion approach and achieves promising empirical results. I'm looking forward to see further experiments on open data sets, computational complexity and different architectures in the future.

**Justification Of The Preliminary Rating:**

The paper addresses a clinically valuable problem with a valid multi-modal fusion approach and achieves promising empirical results. Although certain aspects (such as notation clarity and the “order vs. disorder” concept) need more explanation, they do not overshadow the overall contributions.

**Questions To Address In The Rebuttal:**

How exactly is “order” measured or justified for the decoupled features, and have you considered alternative explicit metrics to quantify it?
Could you clarify how you handle missing or categorical values in the tabular data, and why you view these indicators as a “table modality” rather than simply an input vector?
What is the computational overhead (or GFLOPs) of DeFusion, and how does it compare to simpler fusion techniques such as basic concatenation or addition?
Have you explored any direct comparisons between day 3 and day 5 embryo images to verify whether day 3 can be just as predictive (or nearly so)?

---

> ### Author Response · Authors · 2025-03-07
>
> We sincerely thank the reviewer for the thoughtful feedback. Below, we will answer the general concerns.
> 1. We are sorry for the confusion caused by the lack of clarity in our definition of decoupling. As you say, the decoupling process is a shift from entangled features to disentangled ones. We have verified in Figure 4 that our decoupled features are disentangled by t-SNE and Pearson correlation coefficients.
> 2. We are very sorry for the trouble caused to your reading of the article because of the abbreviation problem. D in Eq. 4 denotes the Decoder, which is composed of three layers of MLP. K in Eq. 3 is the keys in transformer, and K in Eq. 4 is the dimension of the features. N in Eq. 5 is the batch size, and N in section 2.2 is the number of fertility indicators. We will supplement the above details in the official manuscript. In addition, some grammar problems, we will correct.
> 3. In the field of embryo transfer, embryos on day 3 or day 5 are generally selected for transfer. Day 3 embryos are selected because some patients have fewer embryos and are worried about the loss of embryos in the subsequent culture process. Advance transfer can improve the chance of success, and some laboratory culture conditions are limited, the success rate of embryo culture on day 3 is higher. Our study focus on pregnancy prediction for the first 3 days of embryos, so there are no images of embryos developing to day 5 in the dataset. We can compare the method used for pregnancy prediction based on day 5 embryo images and fertility indicators, their AUC result is 0.71, lower than our 0.746 (Enatsu, Noritoshi et al. “A novel system based on artificial intelligence for predicting blastocyst viability and visualizing the explanation.” Reproductive Medicine and Biology 21 (2022): n. pag.).
> 4. For the study of computational complexity, besides dealing with GFLOPs in Table A.2, we also added the training time (in seconds) of a single epoch on 2080Ti GPU with 12G memory, the smaller the index, the better. And Frames Per Second (FPS) on the i7-6850K@3.60GHz CPU, the larger the better. The model, training time(↓), and test FPS(↑) results are as follows:
>
> MMBE: 50, 5.05;
>
> MOAB: 91, 2.69;
>
> SFusion: 65, 3.34;
>
> ConGraph: 68, 3.54;
>
> HMCAT: 79, 2.79;
>
> Ours(DeDusion): 64, 3.51.
>
> Our method is not optimal in computational complexity, which needs to be improved.
> 5. When processing patient fertility indicator data, as shown in Table A.1, because these data are numerical values (have no categorical values) and can be stored in tables, we treat patient data as tabular modality. And for missing values, we use the mean value instead. While MLP or vectors can work well with tabular data, Transformer has stronger feature interaction modeling capabilities for better accuracy as shown in Table 1.
>
> We will supplement the above details in the official manuscript.

---

> > ### Comment · Reviewer_CUmP · 2025-03-11
> >
> > Thank you for answering my questions and considering my concerns in your rebuttal. I have no further questions and will remain with my initial rating.

---

### Official Review · Reviewer_8Sqc · 2025-02-25

**Confidence:** 4
**Preliminary Rating:** 4
**Recommendation:** Poster
**Final Rating:** 4

**Summary:**

This paper proposes a multi-modality framework that utilizes a decoupling method for pregnancy prediction from the IVF-ET process for infertility. The two modalities are temporal embryo images and tabular indicators. For the image modality, a temporal-spatial network is constructed to extract image features. For the tabular data modality, a transformer model is implemented to extract tabular information. The method is tested on a private IVF-ET dataset as well as an eye dataset. The results appear to be very promising.

**Strengths:**

1. It is a good attempt in the IVF-ET area and should be encouraged, as DL methods are lacking in this medical domain.
2. It is a clear and easy-to-follow paper. As a reader, I could grasp the main idea smoothly.
3. The results seem very promising.

**Weaknesses:**

Major Weaknesses
1. Clarity of the methods
 While the authors attempt to illustrate the details of the proposed model, including figures and formulas, some aspects remain unclear in this submission:
   - the fusion classifier: What type of classifier was used?
   - shape of internal features: The representation as $\frac{H}{32}$ is incorrect since H does not necessarily need to be a multiple of 32. Instead, explicitly define H and W in this work and provide the exact numerical shapes.
   - lack of definition for N in Section 2.2.
   - lack of definition for D in Equation 4.
   - lack of definition for $\lambda$ in Section 2.3.

**Detailed Comments:**

Minor Weaknesses
1. paragraph spacing seems incorrect; please check the latex template.
2. figure 1 would be clearer if the two data collection methods were explicitly illustrated.
3. some bold formatting is unnecessary, e.g., in section 3.3. Additionally, in Section 3.4 and the related work, some references are cited incorrectly or awkwardly, e.g., "ViT (**with ViT**)".
4. regarding the related work section, it is recommended to integrate it into the main text.

**Justification Of The Final Rating:**

Thank you for providing the details on the methodology. I have no further questions regarding the content and will maintain my original decision. However, I do have some requests concerning the clarity and formatting of the paper: Please consider revising the notation $``\frac{H}{32}"$, as it does not account for cases where H is not a multiple of 32. Alternatively, explicitly state that H needs to be a multiple of 32. Please double-check your template, as the paragraphs consistently lack indentation at the beginning.

**Justification Of The Preliminary Rating:**

This paper presents a complete study and provides valuable scientific contributions in the domain of IVF-ET. The promising results demonstrate the potential of the proposed model. With improved clarity in the methods section, the paper should be ready.

**Questions To Address In The Rebuttal:**

please refer to the major weakness

**Special Issue:**

No

---

> ### Author Response · Authors · 2025-03-07
>
> We sincerely thank the reviewer for the thoughtful feedback. Below, we will answer the general concerns.
> 1. The fusion classifier is composed of three layers of MLP (Multilayer Perceptron).
> 2. N in Section 2.2 denotes the number of indicators.
> 3. D in Equation 4 denotes the Decoder, which is composed of three layers of MLP.
> 4. λ in Section 2.3 is a hyperparameter in the loss function.
> We will supplement the above details in the official manuscript.

---

> > ### Comment · Reviewer_8Sqc · 2025-03-09
> >
> > Thank you for providing the details on the methodology. I have no further questions regarding the content and will maintain my original decision.
> > However, I do have some requests concerning the clarity and formatting of the paper:
> > Please consider revising the notation $``\frac{H}{32}"$, as it does not account for cases where H is not a multiple of 32. Alternatively, explicitly state that H needs to be a multiple of 32.
> > Please double-check your template, as the paragraphs consistently lack indentation at the beginning.

---

> > > ### Author Response · Authors · 2025-03-12
> > >
> > > Thanks for the reviewer's suggestion, we are indeed using the official 2025 Latex template, and we will add paragraph indentation. And we will convert the formula 512 × (H/32) × (W/32) into C × (H/G) × (W/G), where C, H/G and W/G are respectively the number of channels, height and width of the image after passing through the image backbone network, where G can be evenly divided by H and W.

---

### Author Rebuttal · Authors · 2025-03-07

**Rebuttal:**

Modifications or additions about the manuscript have been marked in red, and the file has been submitted in the supporting material.

**Supporting Material:**

/attachment/8622330cc4eeb11a86e233fbfd7217cf66091b94.zip

---

### Meta-Review · Area_Chair_ew4h · 2025-03-18

**Recommendation:** Accept (Oral)
**Confidence:** 4

**Metareview:**

The reviewers are overall in agreement, highlighting many strengths such as clinical relevance of the problem, completeness of the experiments and promising results (and potential for other interesting research results in the future). After the initial reviews, the authors addressed most of the reviewer concerns, with the reviewers maintaining their initial ratings.

An important problem that was brought up was the lack of an external held-out test set, which the authors corrected by adding an additional out-of-distribution dataset for an external test. I think this is an adequate solution for this work, but it should be emphasized that the single-dataset performances likely do not reflect the expected performance of the method on in-distribution data.

Based on the overall aligned opinions and scores, and the authors' revisions,  I would recommend to accept this paper.